# Sex differences in social support perceived by polymedicated older adults with multimorbidity. MULTIPAP study

**Cristina M. Lozano-Hernández** [1,2,3,4]*, **Juan Antonio López-Rodríguez** [1,3,4,5,6], **Milagros Rico-Blázquez** [1,3,7], **Amaia Calderón-Larrañaga** [3,8,9,10], **Francisca Leiva-Fernández** [3,11,12], **Alexandra Prados-Torres** [3,10], **Isabel del Cura-González** [1,3,5], **MULTIPAP GROUP** [¶]

1 Research Unit, Primary Health Care Management, Madrid, Spain, 2 Interuniversity Doctoral Program in Epidemiology and Public Health, Rey Juan Carlos University, Alcorcon (Madrid), Spain, 3 Research Network on Chronicity, Primary Care and Health Promotion -RICAPPS-(RICORS), Madrid, Spain, 4 Biosanitary Research and Innovation Foundation of Primary Care (FIIBAP), Madrid, Spain, 5 Department of Medical Specialties and Public Health, Faculty of Health Sciences Rey Juan Carlos University, Madrid, Spain, 6 General Ricardos Primary Health Care Centre, Madrid, Spain, 7 Department of Nursing, Faculty of Nursing, Physiotherapy and Podiatry of the Complutense University of Madrid, Madrid, Spain, 8 Joint Action on Chronic Diseases (JA-CHRODIS) European Commission, Brussels, Belgium, 9 Department of Neurobiology, Aging Research Centre, Care Sciences and Society, Karolinska Institute & Stockholm University, Stockholm, Sweden, 10 EpiChron Research Group on Chronic Diseases, Aragon Health Sciences Institute, Miguel Servet University Hospital, Zaragoza, Spain, 11 Multiprofessional Teaching Unit for Family and Community Care Primary Care District Málaga-Guadarhorce, Málaga, Spain, 12 Biomedical Research Institute of Malaga-IBIMA, Andalusian Health Service, Málaga, Spain

¶ Membership of the MULTIPAP GROUP is provided in the Acknowledgments.
* cristinamaria.lozano@salud.madrid.org

**Data Availability Statement:** Regards data sharing, the Aragon Ethics Committee (CEICA, ceica@aragon.es), approved this research without

## Abstract

The beneficial effects of social support on morbidity, mortality, and quality of life are well known. Using the baseline data of the MULTIPAP study (n = 593), an observational, descriptive, cross-sectional study was carried out that analyzed the sex differences in the social support perceived by polymedicated adults aged 65 to 74 years with multimorbidity. The main outcome variable was social support measured through the Duke–UNC-11 Functional Social Support (DUFSS) questionnaire in its two dimensions (confident support and affective support). For both sexes, the perception of functional social support was correlated with being married or partnered and having a higher health-related quality of life utility index. In women, it was correlated with a higher level of education, living alone, and treatment adherence, and in men with higher monthly income, prescribed drugs and fewer diagnosed diseases.

## Introduction

Sociodemographic, environmental, and psychosocial factors can increase the probability of developing health problems [1]. Psychosocial factors such as lack of social support and loneliness have been associated with a higher mortality rate of older adults. In this population group, larger social networks are associated with up to 50% higher probability of survival [2].

considering the option of data sharing. The data contain sensitive clinical information about the patient, so there are ethical and legal restrictions to sharing the data set. The data are part of the MULTIPAP study and can be requested from the Principal Investigators of the project (Alexandra Prados-Torres at sprados.iacs@aragon.es; Daniel Prados-Torres at uand.prados.sspa@juntadeandalucia.es; and Isabel del Cura at isabel.cura@salud.madrid.org). The MULTIPAP Group may establish future collaborations with other groups based on the same data. However, each new project based on these data must be previously submitted to CEICA for approval.

**Funding:** This study was funded by Instituto de Salud Carlos III (ISCIII) [grant references PI15/00276, PI15/00572, PI15/00996, PI18/01812, PI18/01303, PI18/01515, RD16/0001/0004, RD16/0001/0005, RD16/0001/0006] and co-funded by the European Regional Development Fund "A way to shape Europe; Research, Development and Innovation National Plan 2013-2016". CMLH has received a grant from the Fundación para la Investigación e Innovación Biosanitaria de Atención Primaria (FIIBAP) for translation. The funders had no role in study design, data collection and analysis, decision to publish, or preparation of the manuscript.

**Competing interests:** The authors have declared that no competing interests exist.

Although it is not known exactly how social support affects health status, it has both direct and indirect influences. On the one hand, it seems to influence stress and affective changes through the hormonal system and, on the other hand, in behavior, conditioning people's lifestyles [3, 4]. Certain mental health conditions are related to a low perceived social support by the individual, such as anxiety disorders and depression [5, 6]. Likewise, having lower adherence to prescribed treatment and poor health habits has been directly related to having low social support [7–9].

The conceptualization and measurement of social support is a subject of controversy among experts. Not all authors agree on the identification of its dimensions. Norbeck et al. [10] state that the dimensions of social support are so highly correlated that they are difficult to distinguish. Broadly speaking, social support can encompass two dimensions. First, the structural dimension, also called the social network, evaluates the number of links an individual has to others and their interconnections, taking into account the size, frequency of contacts, composition, density, kinship, homogeneity, and strength [11]. Authors such as Hughes et al. and Umberson et al. [12, 13] have suggested that marriage positively influences the healthy behaviors of the individual and therefore their health status. Second, the functional dimension corresponds to the perception of available support flowing through the links of the social network. Dimatteo et al. and Li et al. [7, 14] showed that the network of family and close friends offers more useful social support for the individual than the support of circumstantial friends and acquaintances. One of the most widely used instruments to study perceived functional social support is the Duke-Unk-11, Functional Social Support questionnaire (DUFSS) developed by Broadhead et al. [15]. It originally consisted of 14 items that Broadhead reduced to 11 items in its first validation, and after factor analysis the two-dimensionality of the questionnaire was confirmed. It was identified that on the one hand it measures "confidential support" (the possibility of having people to communicate with) and on the other hand it measures "affective support" (demonstrations of love, affection and empathy).

There are important sex differences in social support and in how women and men perceive it due to sex roles. Traditionally, in men, stereotypes of independence, reflection, aggressiveness, stability, strength, and competitiveness have been emphasized, them being the figure in charge of the defense, production, and economic support of the family, while in women, the stereotypes have been dependency, emotionality, sweetness, instability, weakness, and prudence, them being the family figure linked to care, reproduction, and raising children [16, 17]. According to Cable et al. [18], men report receiving more support from their partners, who are their main source of social support, affirming that marriage has a beneficial effect on psychological well-being and reduces their risk of mortality. Kaplan et al. and Walen et al. [19, 20] have seen that women, on the contrary, more strongly value the support received from their social network of friends, family, and coworkers, resorting to sources outside their partner more frequently than men do [21, 22].

Studies by Berkman and Chen suggest that the relationship between social support and health status is stronger in women [23, 24]. Over their lifetime, women have more comorbidities, multimorbidity, and polymedication and tend to report a lower health-related quality of life (HRQoL) than men of the same age, despite having a lower mortality rate and higher life expectancy [25–28]. Precisely due to their longevity, it has been seen that when women reach a certain age, they have less structural social support than men in the same situation, more often finding themselves living alone [29–32].

The study of the impact of social support, and potential sex differences, in patients with chronic conditions has focused mainly on isolated diseases. Chronicity and longevity tend to generate a need for complex care due to two converging situations in older individuals: multimorbidity, defined as two or more concurrent chronic medical conditions (the threshold of three being more specific for identifying patients with complex health needs) [33], and polypharmacy, defined as the simultaneous consumption of five or more drugs by the same person

[34, 35]. The mean number of chronic problems in the in young senior patients (65–74 years) is estimated to be 2.8 [33, 36, 37], being an understudied age group with an important potential for early intervention.

The main aim of this study is to analyze sex differences in perceived (i.e. functional) social support by polymedicated old adults 65 to 74 years with multimorbidity.

## Materials and methods

An observational, descriptive, cross-sectional study was conducted with an analytical approach using the baseline data of the MULTIPAP study [38]. This intervention study was conducted in 38 health centers in the regions of Andalusia, Aragon, and Madrid (Spain). Patients aged 65–74 years with multimorbidity ($\geq$3 diseases) and polymedication ($\geq$5 different drugs during at least the last 3 months) who had visited their family doctor at least once in the past year and provided written informed consent to participate in the MULTIPAP study [38] were included. Patients residing in nursing homes, with severe mental illness, or with a life expectancy of less than 12 months were excluded. Those patients who met the inclusion criteria were selected by cluster randomised sampling during visits with the 117 participating professionals; five patients per family doctor were enrolled.

The data were collected by previously trained professionals through an interview at the practice. Sociodemographic variables were collected: sex, age, level of education (below primary education, completed primary education, high school, or higher), and professional occupation according to the skill level required by each job through the ISCO-08 [39] (low, medium, or high level). The social class of the household was measured through the CNO-11 [40] (grouped from lowest to highest as VI, V, and I-IV) and monthly household income ($\leq$1050 €/month, 1051–2250 €/month, or $\geq$2251 €/month). The following clinical variables were collected: number of chronic conditions, number of drugs prescribed, diagnosis of depressive disorder and/or anxiety state or disorder, self-reported treatment adherence through the four-item Morisky–Green–Levine Medication Evaluation Questionnaire (MGL MAQ) (0.61 Cronbach's alpha) [41], and HRQoL measured by the EQ-5D-5L [42]. The validated version of the EQ-5D-5L questionnaire for the Spanish population was used [43], which consists of two parts. The first part consists of five questions related to mobility, self-care, daily activities, pain/discomfort, and anxiety/depression. Each one is scored from 1 to 5 points, and from these five questions, a single weighted score is obtained, the Utility Index (EQ-5D-5L Utilities). The scoring for this scale ranges from full health, with a value of 1, to death, with a value of 0, although negative values are allowed. To calculate this index, the algorithm proposed for Spain was used [44]. The second part is a visual analog scale (EQ-5D-5L VAS) that ranges from 0 (worst state) to 100 (best possible health state).

Structural social support was measured through marital status and number of cohabitants in the home. To explore the functional social support, a patient-reported measure was used, namely, the Duke UNC-11 Functional Social Support (DUFSS) questionnaire, which offers a total functional support score and two additional scores referring to the dimensions of confident and affective support [45]. We used the 11-item version with Likert responses of 1 ("much less than I would like") through 5 ("as much as I would like") [46]. The DUFSS questionnaire has been validated in different populations, showing differences in the distribution of the items that make up each of its dimensions. For our study, we chose the validation performed in a noninstitutionalized Spanish population over 65 years of age by Ayala et al. with 0.95 Cronbach's alpha [47]. Its factorial analysis groups items 4, 5, 6, 7, 8, 10, and 11 into the dimension of "confident" support, with a total score of 35, and items 1, 2, 3, and 9, into the dimension of "affective" support, with a total score of 20.

## Analytic plan

The characteristics of study participants and of the social support components were described as frequencies and percentages for qualitative variables and as means ± standard deviations (SD) (normally distributed) or medians and interquartile ranges (IQR) (nonnormally distributed) for quantitative variables. To analyze the associations between the different dimensions of social support and sex, Pearson's chi-squared test was used for qualitative variables and Student's t-test for quantitative variables. Confidence intervals were estimated at 95%.

To study the factors associated with greater functional social support, an explanatory linear regression model was constructed for women and men separately. The dependent variable was functional social support measured through the total score of the DUFSS questionnaire. The independent variables were those that showed statistical significance in the bivariate analysis or were considered relevant in the conceptual framework of the study. Since patients were recruited grouped by clusters (i.e. their family physician), all the estimations were carried out with robust estimators. The analyses were performed with STATA v.14.

The project was approved by the Clinical Research Ethics Committee of Aragon (CEICA) on September 30, 2015. It was favorably evaluated by the Research Ethics Committee of the Province of Malaga on September 25, 2015 and the Central Research Commission of Primary Care of the Community of Madrid on March 16, 2016.

## Results

Of the 593 patients included in the study, 55.8% were women. The mean age of the study population was 69.7 (2.7). Among patients who had not completed their primary studies, the majority were women (64.5% vs. 35.5%, $p < 0.001$). Women were also highly represented among those with lower-skill occupations (83.6% vs. 16.4%, $p < 0.001$) and with the lowest monthly household income (68% vs. 31.8%, $p < 0.001$).

In relation to health status, women had a higher frequency of depressive disorder (82% vs. 18%, $p < 0.001$) and anxiety disorder (77% vs. 23%, $p < 0.001$) than men. No statistically significant difference was found in the number of diagnosed diseases or the number of prescribed drugs. Regarding HRQoL measured by the VAS, women reported a health status 7 points lower than men (69.5 ± 20 vs. 62.5 ± 20.4, $p < 0.001$). Moreover, women more often presented some type of problem in any of the dimensions of HRQoL: mobility (61% vs. 39%, $p = 0.01$); daily activities (73% vs. 27%, $p < 0.001$); pain/discomfort (40.4% vs. 59.6%, $p = 0.001$); and anxiety/depression (29.5% vs. 70.5%, $p < 0.001$). Women had a lower score in utilities than men (0.73 ± 0.2 vs. 0.82 ± 0.2). Table 1 describes the characteristics of the sample according to sex.

**Table 1. Characteristics of the sample according to sex.**

|  | Total n (%) | Men n (%) | Women n (%) |
|---|---|---|---|
|  | **593(100)** | **262(44.2)** | **331 (55.8)** |
| Sociodemographic |  |  |  |
| Age m (SD) | 69.7(2.7) | 69.8(2.6) | 69.7(2.7) |
| Educational level |  |  |  |
| Did not complete primary studies | 279(47.1) | 99(37.8) | 180(54.4) |
| Completed primary studies | 196(33.1) | 82(31.3) | 114(34.4) |
| Bachelor or higher | 118(19.9) | 81(30.9) | 37(11.2) *** |
| Occupation skill level |  |  |  |
| Level 1 | 232(39.1) | 38(14.5) | 194 (58.6) *** |
| Level 2 | 249(42) | 158 (60.3) | 91 (27.5) |

(*Continued*)

**Table 1.** (Continued)

| | Total n (%) | Men n (%) | Women n (%) |
|---|---|---|---|
| | **593(100)** | **262(44.2)** | **331 (55.8)** |
| Level 3 | 80(13.5) | 41(15.7) | 39(11.8) |
| Level 4 | 32(5.4) | 25(9.5) | 7(2.1) |
| Social class of the household | | | |
| VI | 142(24) | 58(22.1) | 84(25.4) |
| V | 217(36.6) | 84(32.1) | 133(40.2) |
| IV-I | 234(39.5) | 120(45.8) | 114(34.4) * |
| Monthly household income | | | |
| < = 1.050 €/month | 170(28.7) | 54(20.6) | 116(35.1) |
| 1.051–2.250 €/month | 342(57.7) | 160(61.1) | 182(54.9) |
| ≥2.251 €/month | 59(10) | 39(14.9) | 20(6.0) *** |
| NS/NC | 22(4) | 9(3.4) | 13(3.9) |
| Clinical | | | |
| Median number of diseases (IQR) | 5(4–7) | 5(4–7) | 5(4–7) |
| Depressive disorder | 110(18.6) | 20(18.2) | 90(81.8) *** |
| Anxiety state or disorder | 88(914.8 | 20(22.7) | 68(77.3) *** |
| Median number of drugs (IQR) | 7(6–9) | 7(5–9) | 7(6–9) |
| MGL MAQ m (SD) | 351(59.2) | 155(44.2) | 196(55.8) |
| HRQoL | | | |
| EQ5D5 L VAS m (SD) | 65.5(20.5) | 69.5(20) | 62.4(20.5) *** |
| EQ5D5 L Utilities m (SD) | 0.77(0.2) | 0.82(0.2) | 0.73(0.2) *** |
| Mobility | | | |
| No problems | 293 (49.4) | 145 (55.3) | 148 (44.7) ** |
| Some type of problem | 300 (50.6) | 117 (44.7) | 183 (55.3) |
| Personal care | | | |
| No problems | 505 (85.2) | 227 (86.6) | 278 (84.0) |
| Some type of problem | 88 (14.8) | 35 (13.4) | 53 (16.0) |
| Daily activities | | | |
| No problems | 411 (69.3) | 213 (81.3) | 198 (59.8) |
| Some type of problem | 182 (30.7) | 49 (18.7) | 133 (40.2) *** |
| Pain/discomfort | | | |
| No problems | 145 (24.5) | 81 (30.9) | 64 (19.3) |
| Some type of problem | 448 (75.6) | 181 (69.1) | 267 (80.7) ** |
| Anxiety/depression | | | |
| No problems | 308 (51.9) | 178 (67.9) | 130 (39.3) *** |
| Some type of problem | 285 (48.1) | 84 (32.1) | 201(60.7) |

Note: m = median; SD = standard deviation; IQR = interquartile range.

* $p < .05$

** $p < .01$

*** $p < .001$

Of the 593 patients, 106 (17.9%) lived alone, of whom 79.3% were women. Men lived in households with three or more cohabitants more frequently than women (59.7% vs. 40.3%, p = 0.009). Sixteen percent of the patients were widowers, and 89.4% of them were women. The mean score of functional social support was 43.7 ± 8.8, with women scoring 2 points lower than men (p = 0.004). Table 2 describes the components of social support by sex. The

**Table 2. Components of social support by sex.**

|  | Total n (%) | Male n (%) | Female n (%) |
|---|---|---|---|
|  | 593(100) | 262(44.2) | 331 (55.8) |
| **Structural Social Support** |  |  |  |
| **Living alone** | 106(17.9) | 22(8.4) | 84(25.4) *** |
| **Living with** |  |  |  |
| 2 people | 368(75.6) | 169(70.4) | 199(80.6) ** |
| ≥ 3 people | 119(24.4) | 71(29.6) | 48(19.4) |
| **Marital status** |  |  |  |
| Single | 23(3.9) | 11(4.2) | 12(3.6) |
| Married | 447(75.4) | 228(87.0) | 219(66.2) |
| Separated | 29(4.9) | 13(5.0) | 16(4.8) |
| Widower | 94(15.9) | 10(3.8) | 84(25.4) *** |
| Functional Social Support (DUFSS) |  |  |  |
| Total score, m (SD) | 43.7(8.8) | 44.9(8.3) | 42.8(9) ** |
| 1st tertile (low) | 190(32) | 70(26.7) | 120(36.3) |
| 2nd tertile (medium) | 191(32.2) | 88(33.6) | 103(31.1) * |
| 3rd tertile (high) | 212(35.8) | 104(39.7) | 108(32.6) |
| "Confident" score, m (SD) | 29.5(5.9) | 30.2(5.7) | 28.9(6.1) ** |
| "Affective" score, m (SD) | 14.2(3.7) | 14.7(3.5) | 13.9(3.7) * |

Note: m = mean; SD = standard deviation; IQR = interquartile range.

* p < .05

** p < .01

*** p < .001

difference in score between the dimensions of functional support for men and women was 1.3 points in the "confident" support dimension and 0.8 points in the "affective" dimension, both scores being lower in women. Table 3 describes the distribution of the DUFSS questionnaire scores by sex. Significant differences by sex appeared regarding the category "Much less/less than I would like" in the items: *I get chances to talk to someone about problems at work or with my housework* (7.3% vs. 14.2%, p = 0.008); *I get chances to talk about money matters* (7.6% vs. 13.9%, p = 0.03); *I get help when I´m sick in bed* (5.7% vs. 13.6%, p = 0.002); *I get help around the house* (29.8% vs. 37.8%, p = 0.04); *I get praise for a good job* (16.0% vs. 25.1%, p = 0.007).

For both sexes, the variable most strongly associated with functional social support was the one referring to utilities in the HRQoL. Table 4 shows the factors associated with functional social support in women and men. For every 1-point increase in the utility score, functional social support increased 11.5 points (95% CI 7.09; 15.85) in women and 9.4 points (95% CI 3.18; 15.59) in men. Being married or partnered was also associated to perceived social support in both women and men, but more strongly in the latter (4.2 points, 95% CI 1.26; 7.07 vs. 3.3 points, 95% CI 0.29; 6.24). The rest of the variables associated to functional social support were different for each sex. In women, the functional social support score increased by 5 points (95% CI 1.91;7.94) in those who had completed high school or higher education; 2 points (95% CI 0.08;3.78) in those adhering to the prescribed treatment; and 5.6 points (95% CI 2.42;8.83) in those who lived alone. In men, the functional social support score increased by 3 points (95% CI 0.74; 5.70) in those with a household income between 1.051–2.250 €/month and 0.5 points (95% CI 0.03; 0.90) for each prescribed drug. Fig 1 shows the magnitude of the association for each of the variables that the final model yielded for both sexes.

**Table 3. DUFSS questionnaire score by sex.**

| | Total n (%) 593(100) | Male n (%) 262(44.2) | Woman n (%) 331 (55.8) | p |
|---|---|---|---|---|
| **Confident Dimension** | | | | |
| Item 4. *I get people who care what happens to me* | | | | |
| Much less/less than I'd like | 49(8.3) | 21(8.0) | 28(8.5) | |
| Neither a lot nor little/almost·as much as I would like | 544(91.7) | 241(92) | 303(91.5) | |
| Item 5. *I get love and affection* | | | | |
| Much less/less than I would like | 59(10) | 21(8.0) | 38(11.5) | |
| Neither a lot nor little/almost·as much as I would like | 534(90.1) | 241(92) | 293(88.5) | |
| Item 6. *I get chances to talk to someone about problems at work or with my housework* | | | | |
| Much less/less than I would like | 66(11.13) | 19(7.3) | 47(14.2) | ** |
| Neither a lot nor little/almost·as much as I would like | 527(88.9) | 243(92.8) | 284(85.8) | |
| Item 7. *I get chances to talk to someone I trust about my personal and family problems* | | | | |
| Much less/less than I would like | 70(11.8) | 25(9.5) | 45(13.9) | |
| Neither a lot nor little/almost·as much as I would like | 523(88.2) | 237(90.5) | 286(86.4) | |
| Item 8. *I get chances to talk about money matters* | | | | |
| Much less/less than I would like | 66(11.1) | 20(7.6) | 46(13.9) | * |
| Neither a lot nor little/almost·as much as I would like | 527(88.9) | 242(92.4) | 285(86.1) | |
| Item 10. *I get useful advice about important things in life* | | | | |
| Much less/less than I would like | 61(10.3) | 25(9.5) | 36(10.9) | |
| Neither a lot nor little/almost·as much as I would like | 532(89.7) | 237(90.5) | 295(89.1) | |
| Item 11. *I get help when I´m sick in bed* | | | | |
| Much less/less than I would like | 60(10.1) | 15(5.7) | 45(13.6) | ** |
| Neither a lot nor little/almost·as much as I would like | 533(89.9) | 247(94.3) | 286(86.4) | |
| **Affective Dimension** | | | | |
| Item 1. *I get visits with friends and relatives* | | | | |
| Much less/less than I would like | 140(23.6) | 55(21) | 85(25.7) | |
| Neither a lot nor little/almost·as much as I would like | 453(79) | 207(45.7) | 246(74.3) | |
| Item 2. *I get help around the house* | | | | |
| Much less/less than I would like | 203(34.2) | 78(29.8) | 125(37.8) | ** |
| Neither a lot nor little/almost·as much as I would like | 390(65.8) | 184(70.2) | 206(62.2) | |
| Item 3. *I get praise for a good job* | | | | |
| Much less/less than I would like | 125(21.1) | 42(16.0) | 83(25.1) | ** |
| Neither a lot nor little/almost/as much as I would like | 468(78.9) | 220(48) | 248(74.9) | |
| Item 9. *I get invitations to go out and do things with other people* | | | | |
| Much less/less than I would like | 116(19.6) | 48(18.3) | 68(20.5) | |
| Neither a lot nor little/almost·as much as I would like | 477(80.4) | 214(81.7) | 263(79.5) | |

Note

* p < .05

** p < .01

*** p < .001

## Discussion and conclusions

There are important sex differences in the social support perceived by polymedicated young-old patients with multimorbidity. These differences must be interpreted bearing in mind the age range studied, i.e. those born in Spain in the 1940s and 1950s, when social differences between men and women were still quite marked [48].

**Table 4. Factors associated with functional social support in women and men.**

| Women | | |
|---|---|---|
| | **Coef. (95% CI)** | **p value** |
| Educational level | | |
| Completed primary studies | 2.33(0.34;4.32) | *0.022* |
| Bachelor or higher | 4.92(1.91;7.94) | *0.001* |
| Adherence, compliance (Morisky-Green) | 1.93(0.08;3.78) | *0.041* |
| Utility index | 11.47(7.09;15.85) | *0.000* |
| Live alone | 5.63(2.42;8.83) | *0.001* |
| Married or partnered | 3.27(0.29;6.24) | *0.031* |
| $R^2$ | | *0.1604* |
| **Men** | | |
| | **Coef. (95% CI)** | **p value** |
| Monthly household income | | |
| 1.051–2.250€/month | 3.22(0.74;5.70) | *0.011* |
| ≥2.251€/month | 2.30(-1.04;5.65) | *0.176* |
| Number of diseases | -0.42(-0.87;0.30) | 0.067 |
| Number of drugs | 0.47(0.03;0.90) | *0.036* |
| Utility index | 9.38(3.18;15.59) | *0.003* |
| Married or partnered | 4.16(1.26;7.07) | *0.005* |
| $R^2$ | | *0.1108* |

The functional social support score reported by women was lower than that reported by men, coinciding with previous studies conducted in similarly aged populations from Spain and Brazil [21, 43]. Being the main source of care and support for others can hinder women's role as a recipient of support from others, which could explain why women, unlike men, have stated that they would like to be better listened to about their problems, hear more praise when they do something well, and get more help when they are sick [29].

The lower social support score perceived by women could also be explained by the fact that social networks are importantly influenced by sex inequalities within social structures. Women with worse educational and occupational level tend to perceive lower social support [25, 45]. In contrast, in men, the perception of social support seems to be related to their income level. Accordingly, a German study found that living in the most socially disadvantaged municipalities was associated with low social support in men, but not in women, which suggests that for men the perception of social support is related to their success in their role as household economic providers, represented in our study by income [49].

For both sexes, being married or partnered increased the perception of social support, observing a stronger association in the case of men, as described by other authors who have found that men's main sources of support are their partners [50]. The increasing feminization of old age has meant that widowhood is a mostly female experience [51]. More and more widowed men and women are living alone, but women do so more often than men, who usually live with someone [52]. Women, when widowed, may feel very supported by social networks that, until then, had not been their main source of support, such as children, other family members, and friends, and may thus perceive greater social support in this new situation [18].

Perceived social support was directly correlated with the utility index of HRQoL in both sexes. In women this relationship was most intense, and they reported worse scores in all dimensions of HRQoL, along the lines of previous research [53]. Different authors have linked

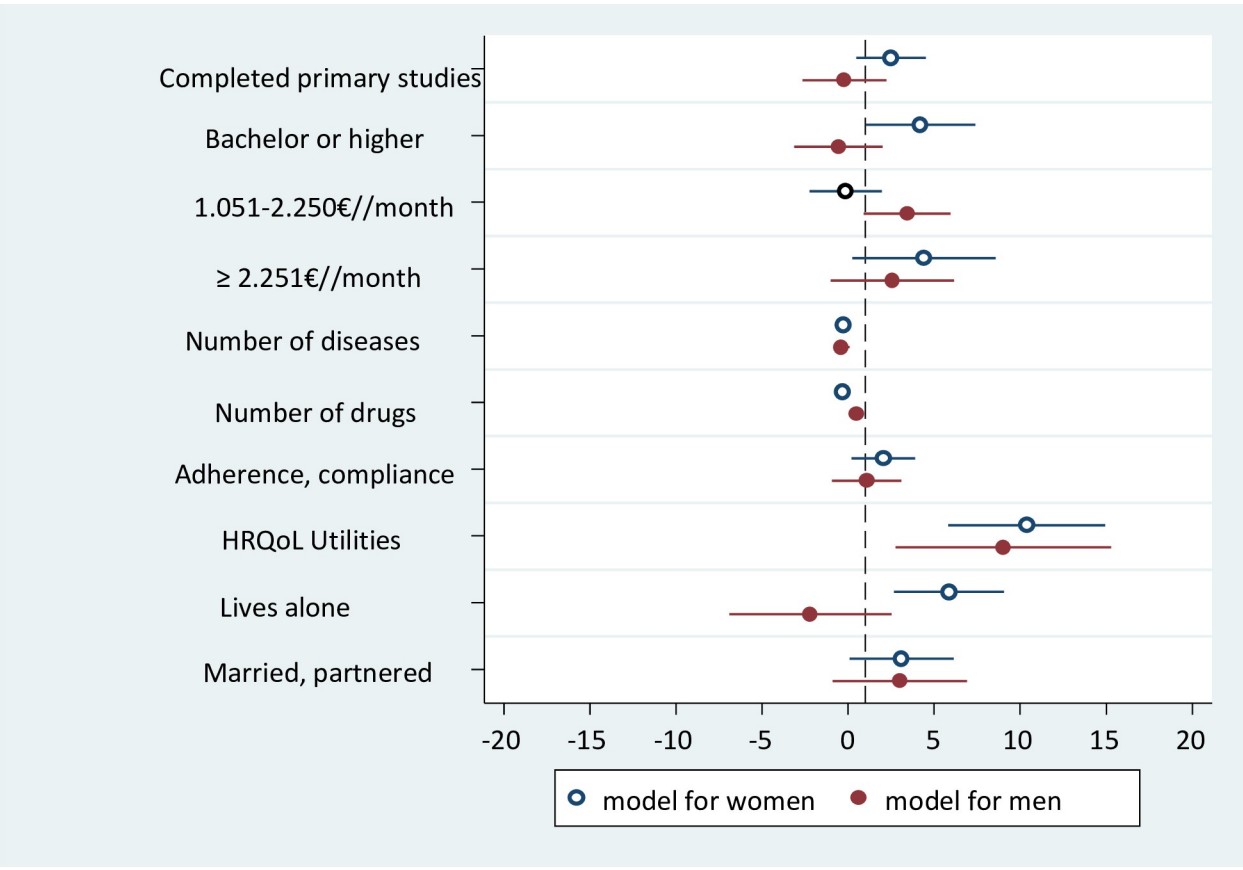

**Fig 1. Differences by sex found in the final model.**

family demands with a worse quality-of-life score, especially in women with lower socioeconomic status [54].

Regarding polypharmacy, in men, better perceived social support was associated with a higher number of prescribed drugs and, in women with better adherence to treatment. This association could be explained by the effectiveness of the treatment, which may improve their health status and symptoms, allowing them to carry out social activities. In the case of women, such an association might be explained by the acceptance and social recognition that they can experience when performing self-care in a socially accepted way [29].

Perceived support is a multidimensional construct subject to different interpretations by experts in the field, for which a consensus is not reached with respect to the dimensions that compose it or how to measure it. Different validations have been carried out on the questionnaire used in this study (DUFSS) in very specific populations, such as caregivers [55], mental health patients [56], or socioeconomically disadvantaged people, and they were mostly women [11, 41]. In our study, we chose to use the most recent validation performed by Ayala et al. [47], who obtained a Cronbach's alpha of 0.94 in noninstitutionalized people aged 60 or over; their population was similar to ours and allows better comparisons with our results.

## Limitations and strengths

The study of social support presents difficulties in relation to conceptualisation and measurement. In this study, structural social support has been studied through a proxy of the social

network such as marital status and the number of cohabitants in the household. This is undoubtedly a limitation, since the structural perspective studies social networks, including all the individual's contacts and providing information on their dimensions, and not only the variables available in this research. However, despite this limitation, this proxy allows us to cover part of the structural social support, following the recommendations of the authors who state that it is more appropriate to study both types of support, functional and structural.

The data used in this study are the baseline data from a randomised clinical trial, MULTI-PAP STUDY [38]. It is a pragmatic cluster randomised controlled clinical trial with 12 months follow-up. The unit of randomisation was the family doctor and the unit of analysis was the patient. Although this was a cross-sectional study, its external validity was increased through systematic random sampling drawn from a representative sampling frame. The sample was drawn from a heterogeneous sample that is representative of the general multimorbid and polymedicated population. . . This was achieved by selecting patients from the health centers, by their family physicians, under clinical practice conditions, giving a pragmatic outlook to the study. The restriction of the sample to the age group of young older adults aged 65 to 74 years makes the sample size difficult, but, at the same time, offers greater knowledge about this group, which is rarely studied on its own but increasingly prominent in our society.

Important sex differences exist in the social support perceived by older multimorbid adults, which should be considered in future public health and health promotion interventions.

## Acknowledgments

To our colleagues from the Research Unit Primary Health Care Management Madrid for their support. To all the professionals from the participant Primary Healthcare Centers. To all patients for their contribution to this research.

## MULTIPAP GROUP:

**Lead authors for the MULTIPAP Study group:** Alexandra Prados Torres (Aragonese Institute of Health Sciences (IACS), IIS Aragón, Miguel Servet University Hospital, Spain) sprados.iacs@aragon.es, Juan Daniel Prados Torres (Multiprofessional Teaching Unit for Family and Community Care Primary Care District Málaga-Guadarhorce. Málaga) juand.prados.sspa@-juntadeandalucia.es, Isabel del Cura (Research unit. Primary Health Care Management Madrid. Spain) isabel.cura@salud.madrid.org.

## Coordinating Committee

José María Abad-Díez (Department of Health, Social Welfare and Family, Government of Aragon), Marta Alcaraz Borrajo (Subdirectorate General of Pharmacy and Health Products), Paula Ara Bardají (Aragonese Institute of Health Sciences (IACS), IIS Aragón, Miguel Servet University Hospital, Spain), Gloria Ariza Cardiel (Research unit. Primary Health Care Management Madrid. Spain), Mercedes Aza-Pascual-Salcedo(Primary Care Department, Aragonese Health Service.), Amaya Azcoaga Lorenzo (Pintores Primary Health Care Centre, Madrid, Spain), Ana Cristina Bandrés-Liso (Primary Care Department, Aragonese Health Service.), Mercedes Clerencia-Sierra (Unit of Social and Health Assessment, Miguel Servet University Hospital, Aragonese Health Service), Nuria García-Agua (Department of Pharmacology, Faculty of Medicine, Malaga University), Luis Gimeno Feliu(San Pablo Primary Health Care Centre, Aragon Health Service, Zaragoza, Spain), Antonio Gimeno-Miguel (Aragonese Institute of Health Sciences (IACS), IIS Aragón, Miguel Servet University Hospital, Spain), Ana I González González(Technical Support Unit, Primary Care Management, Madrid Health Service), Virginia Hernández Santiago(Ninewells Hospital & Medical School,

Dundee, UK), Francisca Leiva Fernández (Multiprofessional Teaching Unit for Family and Community Care Primary Care District Málaga-Guadarhorce. Málaga), Ana Mª López-León (Alhaurín el Grande Health Center, Malaga / Guadalhorce Sanitary District), Juan A López Rodríguez (Research unit. Primary Health Care Management Madrid. Spain), Cristina M Lozano Hernández (Research unit. Primary Health Care Management Madrid. Spain), María Isabel Márquez-Chamizo(Carranque Health Center, Malaga / Guadalhorce Sanitary District.), Alessandra Marengoni(Department of Clinical and Experimental Sciences, University of Brescia, Brescia, Italy), Javier Marta-Moreno(Department of Neurology, University Hospital Miguel Servet, Aragonese Health Service.), Jesús Martín Fernández(Villamanta Primary Health Care Centre, Madrid, Spain), Angel Mataix SanJuan(Subdirección General de Farmacia y Productos Sanitarios), Carmina Mateos-Sancho(Ciudad Jardín Health Center, Malaga / Guadalhorce Sanitary District), Christiane Muth(Institute of General Practice, Johann Wolfgang Goethe University, Frankfurt, Germany), Victoria Pico Soler(Torrero-LaPaz Health Center, Zaragoza, Spain), Beatriz Poblador Plou (Aragonese Institute of Health Sciences (IACS), IIS Aragón, Miguel Servet University Hospital, Spain), Elena Polentinos Castro(Research unit. Primary Health Care Management Madrid. Spain), Antonio Poncel-Falcó (Primary Care Department, Aragonese Health Service.), Ricardo Rodríguez Barrientos (Research unit. Primary Health Care Management Madrid. Spain), José María Ruiz-San-Basilio (Coín Health Center, Malaga / Guadalhorce Sanitary District), Mercedes Rumayor Zarzuelo (6 Centro de Salud Pública de Coslada, Área II Subdirección de Promoción de la Salud y Prevención), Luis Sánchez Perruca (Dirección Sistemas de Información, Gerencia Asistencial de Atención Primaria, Servicio Madrileño de Salud), Teresa Sanz Cuesta (Research unit. Primary Health Care Management Madrid. Spain), Mª Eugenia Tello Bernabé (El naranjo Primary Health Care Centre, Madrid, Spain.), José María Valderas Martínez (University of Exeter Medical School, Exeter, UK. 22Department), Rubén Vázquez-Alarcón (Vera Health Center, AGS Norte de Almería).

## Clinical Investigators in Primary Healthcare Centres (PHC) MULTIPAP GROUP:

**(Andalucía): PCHC Alhaurín el Grande** Javier Martín Izquierdo, Macarena Toro Sainz. **PCHC Carranque Andalucía):** Mª José Fernández Jiménez, Esperanza Mora García, José Manuel Navarro Jiménez.**PCHC Ciudad Jardín Andalucía)::** Deborah Gil Gómez, Leovigildo Ginel Mendoza, Luz Pilar de la Mota Ybancos, Jaime Sasporte Genafo.**PCHC Coín Andalucía)::** Mª José Alcaide Rodríguez, Elena Barceló Garach, Beatriz Caffarena de Arteaga, Mª Dolores Gallego Parrilla, Catalina Sánchez Morales.**PCHC Delicia Andalucía): s:** Mª del Mar Loubet Chasco, Irene Martínez Ríos, Elena Mateo Delgado.**PCHC La Roca Andalucía)::** Esther Martín Aurioles.**PCHC Limonar Andalucía)::** Sylvia Hazañas Ruiz.**PCHC Palmilla Andalucía)::** Nieves Muñoz Escalante.**PCHC Puerta Blanca Andalucía)::** Enrique Leonés Salido, Mª Antonia Máximo Torres, Mª Luisa Moya Rodríguez, Encarnación Peláez Gálvez, José Manuel Ramírez Torres, Cristóbal Trillo Fernández. **PCHC Tiro Pichón Andalucía):** Mª Dolores García Martínez Cañavate, Mª del Mar Gil Mellado, Mª Victoria Muñoz Pradilla. **PCHC Vélez Sur Andalucía):** Mª José Clavijo Peña, José Leiva Fernández, Virginia Castillo Romero.**PCHC Victoria Andalucía):** Rafael Ángel Maqueda, Gloria Aycart Valdés, Miguel Domínguez Santaella, Ana Mª Fernández Vargas, Irene García, Antonia González Rodríguez, Mª Carmen Molina Mendaño, Juana Morales Naranjo, Catalina Moreno Torres, Francisco Serrano Guerra. **Aragón: PCHC Alcorisa** (Alcorisa): Carmen Sánchez Celaya del Pozo.**PCHC Delicias Norte** (Zaragoza): José Ignacio Torrente Garrido, Concepción García Aranda, Marina Pinilla Lafuente, Mª Teresa Delgado Marroquín.**PCHC Picarral** (Zaragoza): Mª José

Gracia Molina, Javier Cuartero Bernal, Mª Victoria Asín Martín, Susana García Domínguez. **PCHC Fuentes de Ebro** (Zaragoza): Carlos Bolea Gorbea.**PCHC Valdefierro** (Zaragoza): Antonio Luis Oto Negre. **PCHC Actur Norte** (Zaragoza): Eugenio Galve Royo, Mª Begoña Abadía Taira.**PCHC Alcañiz** (Alcañiz): José Fernando Tomás Gutiérrez. **PCHC Sagasta— Ruiseñores** (Zaragoza): José Porta Quintana, Valentina Martín Miguel, Esther Mateo de las Heras, Carmen Esteban Algora. **PCHC Ejea** (Ejea de los Caballeros): Mª Teresa Martín Nasarre de Letosa, Elena Gascón del Prim, Noelia Sorinas Delgado, Mª Rosario Sanjuan Cortés. **PCHC Canal Imperial—Venecia** (Zaragoza): Teodoro Corrales Sánchez. **PCHC Canal Imperial—San José Sur** (Zaragoza): Eustaquio Dendarieta Lucas. **PCHC Jaca** (Jaca): Mª del Pilar Mínguez Sorio. Virginia López Cortés.**PCHC Santo Grial** (Huesca): Adolfo Cajal Marzal. **Madrid. PCHC Mendiguchía Carriche** (Leganés): Eduardo Díaz García, Juan Carlos García Álvarez, Francisca García De Blas González, Cristina Guisado Pérez, Alberto López García Franco, Mª Elisa Viñuela Benitez. **PCHC El Greco** (Getafe): Ana Ballarín González, Mª Isabel Ferrer Zapata, Esther Gómez Suarez, Fernanda Morales Ortiz, Lourdes Carolina Peláez Laguno, José Luis Quintana Gómez, Enrique Revilla Pascual. **PCHC Cuzco** (Fuenlabrada): M Ángeles Miguel Abanto.**PCHC El Soto** (Móstoles): Blanca Gutiérrez Teira. **PCHC General Ricardos** (Madrid): Francisco Ramón Abellán López, Carlos Casado Álvaro, Paulino Cubero González, Santiago Manuel Machín Hamalainen, Raquel Mateo Fernández, Mª Eloisa Rogero Blanco, Cesar Sánchez Arce.**PCHC Ibiza** (Madrid): Jorge Olmedo Galindo. **PCHC Las Américas** (Parla): Claudia López Marcos, Soledad Lorenzo Borda, Juan Carlos Moreno Fernández, Belén Muñoz Gómez, Enrique Rodríguez De Mingo. **PCHC Mª Ángeles López** (Leganés): Juan Pedro Calvo Pascual, Margarita Gómez Barroso, Beatriz López Serrano, Mª Paloma Morso Peláez, Julio Sánchez Salvador, Jeannet Dolores Sánchez Yépez, Ana Sosa Alonso. **PCHC Mª Jesús Hereza** (Leganés): Mª del Mar Álvarez Villalba. **PCHC Pavones** (Madrid): Purificación Magán Tapia. **PCHC Pedro Laín Entralgo** (Alcorcón): Mª Angelica Fajardo Alcántara, Mª Canto De Hoyos Alonso, Mª Aránzazu Murciano Antón. **PCHC Pintores** (Parla): Manuel Antonio Alonso Pérez, Ricardo De Felipe Medina, Amaya Nuria López Laguna, Eva Martínez Cid De Rivera, Iliana Serrano Flores, Mª Jesús Sousa Rodríguez. **PCHC Ramón y Cajal** (Alcorcón): Mª Soledad Núñez Isabel, Jesús Mª Redondo Sánchez, Pedro Sánchez Llanos, Lourdes Visedo Campillo.

## Author Contributions

**Conceptualization:** Cristina M. Lozano-Hernández, Milagros Rico-Blázquez, Amaia Calderón-Larrañaga, Francisca Leiva-Fernández.

**Data curation:** Cristina M. Lozano-Hernández, Juan Antonio López-Rodríguez.

**Formal analysis:** Cristina M. Lozano-Hernández, Isabel del Cura-González.

**Funding acquisition:** Cristina M. Lozano-Hernández, Alexandra Prados-Torres, Isabel del Cura-González.

**Investigation:** Cristina M. Lozano-Hernández, Francisca Leiva-Fernández, Alexandra Prados-Torres, Isabel del Cura-González.

**Methodology:** Cristina M. Lozano-Hernández, Juan Antonio López-Rodríguez, Milagros Rico-Blázquez, Isabel del Cura-González.

**Resources:** Cristina M. Lozano-Hernández, Isabel del Cura-González.

**Software:** Cristina M. Lozano-Hernández, Juan Antonio López-Rodríguez.

**Supervision:** Isabel del Cura-González.

**Validation:** Milagros Rico-Blázquez, Amaia Calderón-Larrañaga, Alexandra Prados-Torres, Isabel del Cura-González.

**Visualization:** Amaia Calderón-Larrañaga, Francisca Leiva-Fernández, Alexandra Prados-Torres.

**Writing – original draft:** Cristina M. Lozano-Hernández.

**Writing – review & editing:** Juan Antonio López-Rodríguez, Milagros Rico-Blázquez, Amaia Calderón-Larrañaga, Francisca Leiva-Fernández, Alexandra Prados-Torres, Isabel del Cura-González.

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
