## [Decision Letter · Decision Letter 0]

16 Feb 2022

PONE-D-21-30179

Sex differences in social support perceived by polymedicated older adults with multimorbidity. MULTIPAP Project.

PLOS ONE

Dear Dr. Lozano Hernández,

Thank you for submitting your manuscript to PLOS ONE. After careful consideration, we feel that it has merit but does not fully meet PLOS ONE’s publication criteria as it currently stands. Therefore, we invite you to submit a revised version of the manuscript that addresses the points raised during the review process.

ACADEMIC EDITOR:

Thank you for submitting your work to PLOS ONE and for your patience while it was under review. As you can see, we now have 2 reviews on your submission. Both reviewers were quite positive about this work and highlight their interests in it and strengths of the manuscript. At the same time, though, they both pointed out some important considerations that you should consider addressing. In particular, Reviewer 1, makes some very clear suggestions on improving the link between the conceptual undermining of your approach and the actual data analysis and explanation. 

We look forward to receiving your revised manuscript.

Kind regards,

Andrea Gruneir

Academic Editor

PLOS ONE

https://journals.plos.org/plosone/s/fileid=ba62/PLOSOne_formatting_sample_title_authors_affiliations.pdf".

“This study was funded by Instituto de Salud Carlos III (ISCIII) [grant references PI15/00276, PI15/00572, PI15/00996, PI18/01812, PI18/01303, PI18/01515, RD16/0001/0004, RD16/0001/0005, RD16/0001/0006] and co-funded by the European Regional Development Fund “A way to shape Europe; Research, Development and Innovation National Plan 2013-2016”. CMLH has received a grant from the Fundación para la Investigación e Innovación Biosanitaria de Atención Primaria (FIIBAP) for translation.”

“This study was funded by Instituto de Salud Carlos III (ISCIII) [grant references PI15/00276,

PI15/00572, PI15/00996, PI18/01812, PI18/01303, PI18/01515, RD16/0001/0004,

RD16/0001/0005, RD16/0001/0006] and co-funded by the European Regional Development

Fund “A way to shape Europe; Research, Development and Innovation National Plan 2013-

2016”. CMLH has received a grant from the Fundación para la Investigación e Innovación

Biosanitaria de Atención Primaria (FIIBAP) for translation.”

“This study was funded by Instituto de Salud Carlos III (ISCIII) [grant references PI15/00276, PI15/00572, PI15/00996, PI18/01812, PI18/01303, PI18/01515, RD16/0001/0004, RD16/0001/0005, RD16/0001/0006] and co-funded by the European Regional Development Fund “A way to shape Europe; Research, Development and Innovation National Plan 2013-2016”. CMLH has received a grant from the Fundación para la Investigación e Innovación Biosanitaria de Atención Primaria (FIIBAP) for translation.”

6. Please note that in order to use the direct billing option the corresponding author must be affiliated with the chosen institute. Please either amend your manuscript to change the affiliation or corresponding author, or email us at plosone@plos.org with a request to remove this option.

7. One of the noted authors is a group or consortium [MULTIPAP group]. In addition to naming the author group, please list the individual authors and affiliations within this group in the acknowledgments section of your manuscript. Please also indicate clearly a lead author for this group along with a contact email address.

8. Please include your tables as part of your main manuscript and remove the individual files. Please note that supplementary tables (should remain/ be uploaded) as separate ""supporting information"".

Reviewers' comments:

Reviewer's Responses to Questions

**Comments to the Author**

1. Is the manuscript technically sound, and do the data support the conclusions?

Reviewer #1: Yes

Reviewer #2: Yes

2. Has the statistical analysis been performed appropriately and rigorously? 

Reviewer #1: Yes

Reviewer #2: Yes

3. Have the authors made all data underlying the findings in their manuscript fully available?

Reviewer #1: Yes

Reviewer #2: Yes

4. Is the manuscript presented in an intelligible fashion and written in standard English?

Reviewer #1: Yes

Reviewer #2: Yes

5. Review Comments to the Author

Reviewer #1: This manuscript uses survey data from Spanish health centers to assess the sex differences in social support among older polymedicated adults. Overall, it is fairly well written and easy to follow. Please see below for my comments.

1. For the most part, the authors do well to use politically correct language when referring to older adults. However, on line 45, per the Gerontological Society of America’s policy on language regarding the older population, I suggest replacing the term ‘eldery people’ with a more neutral term (e.g., older adults, older people). See for (Lundebjerg et al., 2017) for details on this policy.

CITED SOURCE: Lundebjerg, Nancy E., Daniel E. Trucil, Emily C. Hammond, and William B. Applegate. 2017. “When It Comes to Older Adults, Language Matters: Journal of the American Geriatrics Society Adopts Modified American Medical Association Style.” Journal of the American Geriatrics Society 65(7):1386–88. doi: 10.1111/jgs.14941.

2. The authors do well to distinguish between social networks and social support (lines 56-65). However, after introducing the concept of networks (including size, density, homogeneity, etc) in the Introduction, it was a little disappointing to see structural social support later operationalized simply as marital status and number of cohabitants in the home (lines 122-123). While these measures are often indicative of social support, their mere presence does not automatically guarantee social support. For instance, poor marriages might actually cause more strain than support (de Jong Gierveld et al. 2009). I realize that data limitations are always going to be an issue, but it seems that it is at least acknowledging the lack of social network data and marital quality as limitations in this study.

CITED SOURCE: Jenny de Jong Gierveld, Marjolein Broese van Groenou, Adriaan W. Hoogendoorn, Johannes H. Smit, Quality of Marriages in Later Life and Emotional and Social Loneliness, The Journals of Gerontology: Series B, Volume 64B, Issue 4, July 2009, Pages 497–506, https://doi.org/10.1093/geronb/gbn043

3. In the Materials and Methods section, the authors choose to separate functional support into ‘confident support” and “affective support” (lines 131-133). If they are going to make this distinction methodologically, it should be preceded with conceptual justification in the Introduction section.

4. I don’t understand the values given in lines 175-178). They do not seem to match those in Table 3. The bivariate comparisons given in the text do not make sense to me because they are not dichotomous outcomes. Rather there appears to be three response categories to each outcome.

5. I am confused by the sentence on line 215 that reads “The increasing feminization of old age has meant that widowhood is a mostly female experience.” What does this mean? Are the authors trying to say that women are more likely to experience widowhood because they tend to live longer than men?

6. The sentence on lines 241-243 the authors say that their heterogeneous sample “is representative of the general multimorbid and polymedicated population, increasing its external validity.” But it seems to me that the external validity depends more on the sampling methodology (i.e., random sampling drawn from a representative sampling frame) rather than the sample demographics. It is difficult to tell if participants were randomly selected or if a convenience sampling method was used instead. Perhaps more importantly, however, the participants appear to have first been contacted in health centers, which—if true—would not make it a representative sample of the general population since only those who went to the health centers were eligible for being selected into the study. These distinctions should be noted in the manuscript.

Reviewer #2: I have just read the manuscript entitled "Sex differences in social support perceived by polymedicated older adults with multimorbidity. MULTIPAP Project", an interesting work exploring determinants of social support perceived by sex. The study is focused on polymedicated older adults with multimorbidity. After reading the manuscript, I have the following comments:

- The article is focused in older adults, although only in those aged 65-74 years. Why people older than 75 were not considered in the sample?

- All the instruments used in this study have been previously validated in other samples. I wonder whether the authors could provide data on internal consistency (e.g., Cronbach’s alpha for unidimensional scales) according to the sample considered in this study.

- In Table 4, the 95% CI associated to “Number of drugs” in men cannot provide a significant p-value. This would be checked and also revised in the Results section in the main text.

- The authors have found an association between lower perception of social support and multimorbidity in men. This result is not observed in women and should be discussed.

- Potential limitations of this study should be highlighted in the Discussion section.

Minor comments:

- Line 101: "Random sampling". Which type of random sampling was used?

- Lines 141-142: “Explanatory linear regression models”. Why explanatory?

- Lines 190-192: “In contrast, a higher number of diseases was associated with a lower social support score; it fell by 0.4 points (95% CI -0.87–0.30) for each disease”. Please revise the use of hyphen in confidence intervals when negative values are reported. On the other hand, it is not a significant result.

6. PLOS authors have the option to publish the peer review history of their article (what does this mean?). If published, this will include your full peer review and any attached files.

Reviewer #1: No

Reviewer #2: No

---

## [Author Response · Author response to Decision Letter 0]

31 Mar 2022

We are very grateful for your evaluation of our manuscript entitled "Sex differences in social support perceived by polymedicated older adults with multimorbidity. MULTIPAP Study" 

We appreciate the comments of the reviewers and editors and have made the suggested changes. We have attached the reviewers' response.

We hope that these changes will contribute to improve the quality of the manuscript and the interest of potential readers.

On behalf of the research team,

Best regards,

Cristina María Lozano Hernández

---

## [Decision Letter · Decision Letter 1]

26 Apr 2022

Sex differences in social support perceived by polymedicated older adults with multimorbidity. MULTIPAP Study.

PONE-D-21-30179R1

Dear Dr. Lozano Hernández,

We’re pleased to inform you that your manuscript has been judged scientifically suitable for publication and will be formally accepted for publication once it meets all outstanding technical requirements.

Kind regards,

Thomas Penzel

Academic Editor

PLOS ONE

Additional Editor Comments (optional):

Reviewers' comments:

Reviewer's Responses to Questions

**Comments to the Author**

1. If the authors have adequately addressed your comments raised in a previous round of review and you feel that this manuscript is now acceptable for publication, you may indicate that here to bypass the “Comments to the Author” section, enter your conflict of interest statement in the “Confidential to Editor” section, and submit your "Accept" recommendation.

Reviewer #1: All comments have been addressed

Reviewer #2: All comments have been addressed

2. Is the manuscript technically sound, and do the data support the conclusions?

Reviewer #1: Yes

Reviewer #2: Yes

3. Has the statistical analysis been performed appropriately and rigorously? 

Reviewer #1: Yes

Reviewer #2: Yes

4. Have the authors made all data underlying the findings in their manuscript fully available?

Reviewer #1: Yes

Reviewer #2: Yes

5. Is the manuscript presented in an intelligible fashion and written in standard English?

Reviewer #1: Yes

Reviewer #2: Yes

6. Review Comments to the Author

Reviewer #1: (No Response)

Reviewer #2: I have read the last version of the manuscript entitled "Sex differences in social support perceived by polymedicated older adults with multimorbidity. MULTIPAP Study" and the authors have adequately addressed all my previous comments.

7. PLOS authors have the option to publish the peer review history of their article (what does this mean?). If published, this will include your full peer review and any attached files.

Reviewer #1: No

Reviewer #2: No

---

## [Editor Report · Acceptance letter]

28 Apr 2022

PONE-D-21-30179R1 

Sex differences in social support perceived by polymedicated older adults with multimorbidity. MULTIPAP Study. 

Dear Dr. Lozano-Hernández:

I'm pleased to inform you that your manuscript has been deemed suitable for publication in PLOS ONE. Congratulations! Your manuscript is now with our production department. 

Kind regards, 

on behalf of

Dr. Thomas Penzel 

Academic Editor

PLOS ONE